# Determining hexavalent chromium transport properties in alkaline nuclear waste using nuclear magnetic resonance spectroscopy

Trent R. Graham [1] ✉, Ashley R. Kennedy [1,2], Jacob Morton[1], Jacob G. Reynolds[3] & Carolyn I. Pearce [1,4]

This study focuses on the transport properties of hexavalent chromium, specifically the chromate anion, to improve predictive models and environmental remediation strategies for Cr(VI) migration. Using [53]Cr Nuclear Magnetic Resonance (NMR) spectroscopy, the research quantifies chromate in multicomponent electrolytes replicating nuclear waste conditions at the Hanford Site in Washington State. The consistency of the [53]Cr NMR signal integral with chromate concentration, despite varying matrix compositions, establishes it as a reliable concentration indicator. The transport properties of chromate in an alkaline solution were assessed using relaxation-based measurements via saturation recovery and Carr-Purcell-Meiboom-Gill experiments, determining spin-lattice and spin-spin relaxation times. These measurements, combined with the Bloembergen-Purcell-Pound equation, helped estimate the rotational correlation time and the [53]Cr self-diffusion coefficient using Stokes-Einstein-Debye and Stokes-Einstein equations. Direct measurements were obtained through pulsed field gradient stimulated echo [53]Cr NMR spectroscopy. Monte Carlo simulations further estimated uncertainty propagation. The results enhance comprehension of chromate transport and highlight prospects for identifying transport properties of NMR-active nuclei, traditionally considered unreachable.

Hexavalent chromium [Cr(VI)] pollution poses a significant environmental hazard in industrial effluents and wastewater[1]. This issue is relevant at sites of historical nuclear activities, such as the Hanford Site in Washington State, where it is produced from oxidative leaching of trivalent chromium[2,3]. Along with Hanford, other places like the Rivera Site in Switzerland[4] and New Caledonia, South Pacific contain Cr(VI) contamination in ground water[5]. Predictive insights into Cr(VI) transport properties of colloids and ions within environmental matrices requires detailed characterization of its chemical environment, interactions, and intrinsic mobility[6–8]. The transport of Cr(VI) in groundwater systems is influenced by complex interactions at redox interfaces, especially in environments with iron-bearing minerals[9]. The redox potential, pH, presence of natural organic matter, and microbial activity can significantly influence these transport processes[10–14]. Moreover, the redox chemistry of Cr(VI) in soil is coupled to that of common species such as nitrate[15], iron (hydr)oxides[16,17], and manganese oxides[18], as well as in

some circumstances to uranium and technetium[19–21], which will impact Cr(VI) transport behavior in complex systems. According to the Pourbaix diagram, the chromate anion ($CrO_4^{2-}$) is the primary species in aqueous solutions at high alkalinities and redox potentials (Eh) above 0 V, which are representative of the alkaline conditions in radioactive waste stored in tanks at the Hanford Site[22].

Understanding the characteristics and mobility of Cr(VI) under these conditions requires the use of specialized experimental methodologies. Many techniques, such as Inductively Coupled Plasma Mass Spectrometry (ICP-MS) can quantify Cr(VI) concentrations, but it is highly sensitive for detecting trace elements and results are affected by matrix interferences. X-ray absorption spectroscopy (XAS) can also be used to determine Cr oxidation state and coordination environment, but it is not suitable for overall Cr quantification, and complex environmental samples require access to synchrotron radiation sources[23]. Electrochemical methods

[1]Pacific Northwest National Laboratory, Richland, WA, USA. [2]Savannah River National Laboratory, Aiken, SC, USA. [3]Central Plateau Cleanup Company, LLC, Richland, WA, USA. [4]Department of Crop and Soil Sciences, Washington State University, Pullman, WA, USA. ✉e-mail: trent.graham@pnnl.gov

facilitate in-situ measurements but are often affected by interference from other redox-active species. Given these challenges, there is a need for complementary techniques that can provide quantitative information on Cr concentration and speciation, along with molecular scale transport dynamics to supplement existing macroscale techniques, such as those using isotope tracers[24] and analyses of effluents from break through columns[25].

A less commonly used approach is $^{53}$Cr Nuclear Magnetic Resonance (NMR) spectroscopy. $^{53}$Cr NMR spectroscopy was described as early as 1969[26], and benefits from a degree of specificity to Cr(VI) and Cr(0) as trivalent chromium is paramagnetic and not directly observed[27]. However, the $^{53}$Cr nucleus has a low gyromagnetic ratio ($-1.51520107$ rad/(T·s)) and Larmor frequencies (e.g. 28.262 MHz at 11.74 T), resulting in low receptivity (0.50765 relative to $^{13}$C, when $^{53}$Cr is at natural abundance of 9.501%)[28]. Alongside the quadrupolar spin 3/2 properties of $^{53}$Cr, this leads to challenges in measuring asymmetric environments such as those of dissolved dichromate salts, although with some exceptions[29]. The low gyromagnetic ratio and quadrupolar effects on relaxation kinetics are especially detrimental for diffusion measurements via pulsed field gradient techniques[30–33].

The present study demonstrates the robustness and reliability of quantifying $CrO_4^{2-}$ concentration in multicomponent electrolytes that simulate Hanford tank waste, despite matrix variations. The transport properties of $CrO_4^{2-}$ are then investigated in an idealized alkaline aqueous solution of 1.47 m KOH and 1.75 m $K_2CrO_4$. The concentrated solution was characterized to estimate Cr transport properties using saturation recovery and Carr-Purcell-Meiboom-Gill (CPMG) experiments to determine $T_1$ and $T_2$ coefficients. By leveraging $T_1$ and $T_2$ measurements, diffusion coefficients can be predicted through the rotational correlation time ($\tau_c$) and the Stokes-Einstein-Debye and Stokes-Einstein equations. These predictions are then compared with direct measurements obtained via PFGSTE $^{53}$Cr NMR spectroscopy, supplemented by Monte Carlo simulations to analyze the uncertainty. Monte Carlo simulations have been used to estimate error propagation in comparable applications such as magnetic resonance imaging[34,35] along with quantitative NMR spectroscopy of small molecules, macromolecules, and polymers[36–38], in addition to instrument development[39]. Applying Monte Carlo methodologies to $^{53}$Cr NMR spectroscopy facilitated inspection of the errors and uncertainties in relaxation time measurements, as well as their propagation to diffusion coefficients, both through analysis of relaxation times and also through direct measurement with PFGSTE NMR spectroscopy. These insights will enable more accurate modeling of Cr transport, which is necessary to develop effective environmental remediation strategies. This work also provides a foundation to apply PFGSTE NMR spectroscopy to other unfavorable NMR-active nuclei for description of transport properties in systems relevant to energy sciences.

## Results and discussion

The results and discussion are organized as follows: First, we estimate the limit of detection (LOD) and linearity with respect to concentration of $CrO_4^{2-}$ in multicomponent Hanford simulants via analysis of single pulse direct excitation $^{53}$Cr NMR spectra. Given that $CrO_4^{2-}$ is dilute (<20 mM) in these simulants, further experiments to estimate transport properties were conducted with concentrated $CrO_4^{2-}$ in KOH. While sodium is the dominant cation in Hanford tank waste, KOH was used in place of NaOH to mitigate the effects of the highly viscous NaOH solutions while retaining $CrO_4^{2-}$ as the dominant Cr(VI) species in alkaline solutions.

Experiments were conducted to determine the LOD in five industrially relevant simulants targeting supernatant stored in the Hanford Site's 200 West Area. As shown in Fig. 1A, based on historical data, the key analytes in the mixture were $CrO_4^{2-}$, $Al(OH)_4^-$, $C_2O_4^{2-}$, $CO_3^{2-}$, $SO_4^{2-}$, $PO_4^{3-}$, $NO_2^-$, $NO_3^-$, $Cl^-$, $F^-$, $OH^-$, $Ca^{2+}$, $Cs^+$, $K^+$, $Na^+$, $Sr^{2+}$, $CH_3CO_2^-$, and $CHO_2^-$. Detailed elsewhere[40], the chemistry of the simulants was characterized using several techniques, including ICP-OES, ICP-MS, ion chromatography, and acid titration, along with measurements of liquid phase density and viscosity. The hydroxide concentration in the solutions was measured by titration as 2.04, 0.731, 0.408, 1.01, and 0.672 M $OH^-$ for S1 through S5,

respectively. These solutions provide a series of samples in which the sensitivity of $^{53}$Cr can be determined at representative compositions relevant to nuclear waste processing. Table S3 in the *Supplementary Information* tabulates the concentrations of the constituents in the simulants.

Figure 1B shows that the $^{53}$Cr resonance can still be observed even in dilute solutions under 10 mM Cr. The resonance appears as a Lorentzian line near 0.7 ppm (where 0 ppm is referenced to the resonance of 1.47 m KOH and 1.75 m $K_2CrO_4$). With 24 h of data collection, the signal has fair signal-to-noise ratios, and the integration of this signal yielded a high ($R^2$) value of 0.997 when fitted to a linear function dependent on the concentration of $CrO_4^{2-}$ as shown in Fig. 1C. The linear relationship between the integral and Cr(VI) concentration, despite the variable composition of the multicomponent concentrated electrolyte, indicates that this is a reliable approach to quantify Cr(VI) concentration under diverse, process relevant conditions. Given the lengthy acquisition time required for obtaining low signal-to-noise spectra with single pulse direct excitation $^{53}$Cr NMR experiments, further NMR experiments were conducted with 1.47 m KOH and 1.75 m $K_2CrO_4$. The high pH restricts formation of $Cr_2O_7^{2-}$[22,26], and using KOH instead of NaOH limits the increases in dynamic viscosity[41]. As shown in Fig. S1 of the *Supplementary Information*, the speciation of $CrO_4^{2-}$ in the KOH solution was confirmed with $^{17}$O NMR spectroscopy, which showed the absence of $^{17}$O NMR resonances associated with $Cr_2O_7^{2-}$ and the presence of the resonance associated with $CrO_4^{2-}$[42].

Saturation recovery and CPMG experiments were used to quantify the $T_1$ and $T_2$ relaxation times in concentrated $CrO_4^{2-}$ in KOH solution followed by the prediction of diffusivity coefficients using the Bloembergen-Purcell-Pound (BPP), Stokes-Einstein, and Stokes-Einstein-Debye equations[43,44].

To measure $T_1$, saturation recovery experiments were performed, which involve saturating the spins with a train of pulses, followed by a variable recovery delay before the observation pulse[45]. Note that in lieu of inversion recovery, saturation recovery experiments were chosen to reduce overall experimental time and allow for a more extensive set of PFGSTE NMR experiments, as saturation recovery does not require the $T_1$ value to be known a priori. Figure 2A shows the saturation recovery spectra acquired at 20 °C, annotated with the time delay between the saturation train and the excitation pulse. In Fig. 2C, the data are fitted with a mono-exponential curve, and Fig. 2E shows the temperature dependence of the relaxation times, which exhibited Arrhenius behavior. Regarding the noise, in Fig. 2C, the data at 60 °C have the highest normalized uncertainty of about 5%, followed by lower uncertainties at other temperatures. While we cannot completely isolate the contributions to noise, the magnitude of these uncertainties does not vary systematically with temperature or with d20 (time delay in saturation recovery experiments).

Next, to measure $T_2$, CPMG experiments were conducted, which involve a series of 180° refocusing pulses following an initial 90° pulse[46]. This sequence echoes the spins multiple times, effectively refocusing dephasing spins due to magnetic field inhomogeneities and thus measuring the spin-spin relaxation time ($T_2$). Figure 2B shows the spectra acquired at 20 °C. Figure 2D then illustrates the analysis of the CPMG experiments to determine $T_2$ coefficients. A mono-exponential equation was used to fit the CPMG experiments. Lastly, Fig. 2E shows the temperature dependence of the $T_2$ relaxation times, highlighting Arrhenius behavior and indicating the sensitivity of the spin-spin relaxation processes to temperature variations.

The relationship between $T_1$ and $T_2$ can then be used to estimate the rotational correlation time, ($\tau_c$), for spin-3/2 nuclei (Eq. 1)[44,47,48], which describes the characteristic time scale of molecular motion. As shown in Fig. 2F, $\tau_c$ was approximately constant across the temperature range, and $\tau_c$ was then used to calculate the self-diffusion coefficient ($D_T$) of $^{53}$Cr using the Stokes-Einstein-Debye and Stokes-Einstein Equations (Eqs. 2–4)[43].

$$\frac{T_2}{2T_1} = \frac{\frac{1}{1+\omega_0^2\tau_c^2} + \frac{4}{1+4\omega_0^2\tau_c^2}}{3 + \frac{5}{1+\omega_0^2\tau_c^2} + \frac{2}{1+4\omega_0^2\tau_c^2}} \tag{1}$$

 

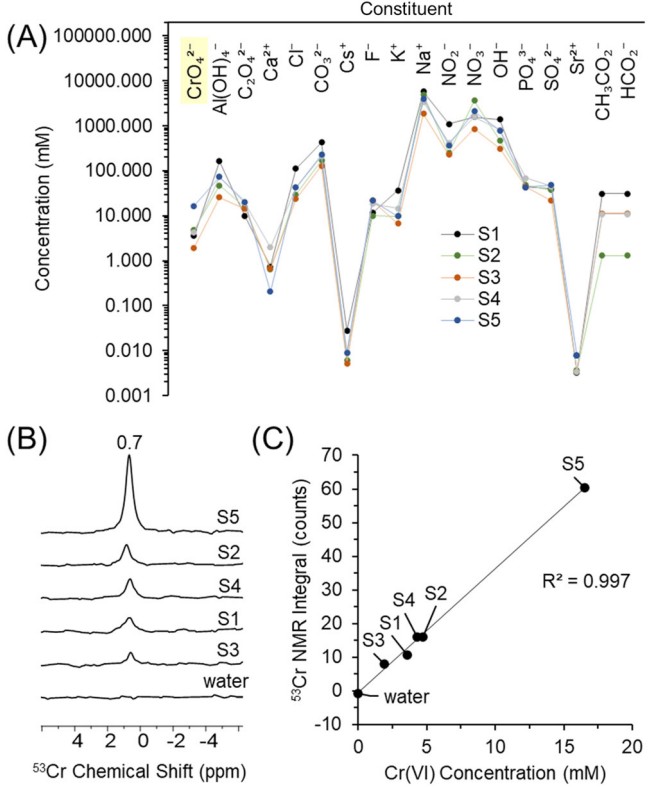

**Fig. 1 | Analysis of Hanford tank waste simulants. A** Concentrations of species of interest in five different Hanford tank waste simulants to show trends in composition. The lines in the parallel coordinates diagram are drawn only to guide the eyes. **B** $^{53}Cr$ resonance of $CrO_4^{2-}$ in the five different Hanford tank waste simulants. Spectra correspond to ~1 day of signal averaging. **C** Integral of $^{53}Cr$ NMR data in (**B**) versus Cr(VI) concentration determined by ICP-OES with a linear fit to the integrated $^{53}Cr$ NMR resonance. Note the units of concentration are millimoles/L (mM).

In Eq. 1, $\omega_0$ is the Larmor Frequency of $^{53}Cr$ at a magnetic field strength of 11.74 T and $\tau_c$ is the correlation time.

$$D_r = \frac{1}{6\tau_c} \tag{2}$$

$$D_r = \frac{K_B T}{8\pi\eta R_h^3} \tag{3}$$

$$D_T = \frac{K_B T}{6\pi\eta R_h} \tag{4}$$

In Eqs. 2–4, $D_r$ is the rotational diffusion coefficient, $K_B$ is the Boltzmann coefficient, $T$ is temperature, $\eta$ is the experimentally measured[49] dynamic viscosity, $R_h$ is hydrodynamic radius, and $D_T$ is the transverse diffusivity coefficient. These diffusion coefficients predicted with $T_1$ and $T_2$ were then compared to direct measurements of diffusion at 20 °C acquired with $^{53}Cr$ PFGSTE NMR spectroscopy.

Direct measurement $D_T$ was then acquired with $^{53}Cr$ PFGSTE NMR spectroscopy. Pioneered by Stejskal and Tanner in the 1960s, PFG NMR spectroscopy advanced the study of molecular diffusion by enabling precise NMR measurement of self-diffusion coefficients through the Stejskal-Tanner equation (Eqs. 5–6)[50].

$$S(b) = S_0 \exp(-bD_T) \tag{5}$$

$$b = \gamma^2 G^2 \delta^2 (\Delta - \delta/3) \tag{6}$$

In Eqs. 5–6, $S$ is the echo intensity, $b$ is the attenuation coefficient, $S_0$ is the initial echo intensity, $\gamma$ is the gyromagnetic ratio of $^{53}Cr$, $\delta$ is the length of the gradient pulse, $\Delta$ is the time in which the molecules diffuse between gradient pulses, and $G$ is the amplitude of the gradient pulse. Note that in practice, fitting is done to the modified Stejskal-Tanner equation with adjustments for smooth-square gradients, bipolar pulses, and convection compensation, using Topspin software to scale gradient steps and ensure adherence to the canonical form[51]. Due to the small gyromagnetic ratio of $^{53}Cr$, stronger gradients are needed to spatially resolve the spins. Whereas $\Delta$ can also be typically increased to improve the degree of signal attenuation, this is constrained by signal loss via $T_1$ and $T_2$ processes. By using a PFG sequence constructed around a stimulated echo, attenuation losses due to the shorter $T_2$ relative to $T_1$ are minimized. Hence, the PFGSTE NMR experiment was selected to determine diffusion coefficients in this study.

Figure 3A presents the PFGTSE NMR spectra of these direct measurements. Attenuation is observed, but only a signal reduction to approximately 80% is obtained. In Fig. 3B, the data is analyzed in a Stejskal Tanner plot which such that the decay as a function of $b$ is observable. The exponential fit appears quasi-linear across the measurable range of b due to the limited dynamic range of signal attenuation. Lastly, Fig. 3C shows the PFGSTE NMR experiments alongside the diffusion coefficients determined via analysis of $T_1$ and $T_2$ coefficients. The direct measurement of the self-diffusion coefficient of $^{53}Cr$ using PFGSTE NMR, despite inherent challenges related to the small gyromagnetic ratio and experimental constraints, demonstrates the feasibility of obtaining diffusion data. The fair agreement between direct measurements and relaxation-based estimates validates the methodological approach and highlights the effectiveness of combining multiple techniques to achieve comprehensive diffusivity characterization. Given the low concentration of $CrO_4^{2-}$ in Hanford tank waste, diffusivity estimates via relaxometry in tandem with isotopic enrichment may potentiate further description, which is outside the scope of the current work. Due to the considerable noise in the PFGSTE NMR results, Monte Carlo simulations were performed to estimate the uncertainty associated with the diffusion coefficient, with a focus on whether the uncertainty was symmetric and the degree to which noise characteristics affect the accuracy of the diffusion coefficients.

Monte Carlo simulations were used to model and understand the impact of uncertainty and variability propagating from the $T_1$ and $T_2$ measurements to the $D_T$ via the Bloembergen-Purcell-Pound (BPP), Stokes-Einstein, and Stokes-Einstein-Debye equations. Ensembles of simulated data were generated from the fit by adding Gaussian noise in the frequency domain with variance matching the residuals between the observed data and the fitted model. The resulting ensemble was then analyzed in the same manner as the original data to produce distributions of fit parameters $T_1$, $T_2$, $\tau_c$, and $D_T$ that facilitated inspection of asymmetric uncertainty.

Figure 4A-B show 20 example simulations, in addition to the original data for the saturation recovery experiment and the CPMG experiment, along with distributions of the $T_1$ and $T_2$ coefficients obtained from over 10,000 simulations as shown in Fig. 4D-E. For each Monte Carlo Simulation, the resulting $T_1$ and $T_2$ were analyzed via BPP to determine the correlation time, shown in Fig. 4F, and lastly these rotational correlation coefficients were applied to Stokes-Einstein-Debye and Stokes Einstein equations to determine the diffusion coefficient in Fig. 4G. Similarly, 20 example simulations of the PFGSTE NMR experiment are shown in Fig. 4C, with the corresponding distribution of $D_t$ from 14,000 simulations shown in Fig. 4H. The distributions of $T_1$, $T_2$, $\tau_c$, and $D_T$ were all characterized by both Gaussian and Skewed Gaussian distributions, because there was evidence of a limited amount of asymmetry relative to an ideal Gaussian distributions for $T_1$, $T_2$, $\tau_c$ and the resulting $D_T$ acquired via analysis of the relaxation times. The significance of these asymmetrical nature of the skewed Gaussian is that the center of mass and standard deviation of the skewed distributions can vary

**Fig. 2 | NMR Relaxometry of 1.47 m KOH and 1.75 m K$_2$CrO$_4$. A** $^{53}$Cr saturation recovery spectra at 20 °C. **B** $^{53}$Cr CPMG spectra at 20 °C. **C** Data fit to determine $T_1$ and (**D**) $T_2$ as a function of temperature. **E** Diagram to show Arrhenius relationship to determine the temperature dependence of the $T_1$, $T_2$ and (**F**) $\tau_c$. The error bars (+/- σ) in this figure were determined via analysis of Monte Carlo simulations as later described. Tables of $T_1$, $T_2$ and $\tau_c$ are provided in Tables S4–S6 of the *Supplementary Information*. Note, the units of concentration are in molality.

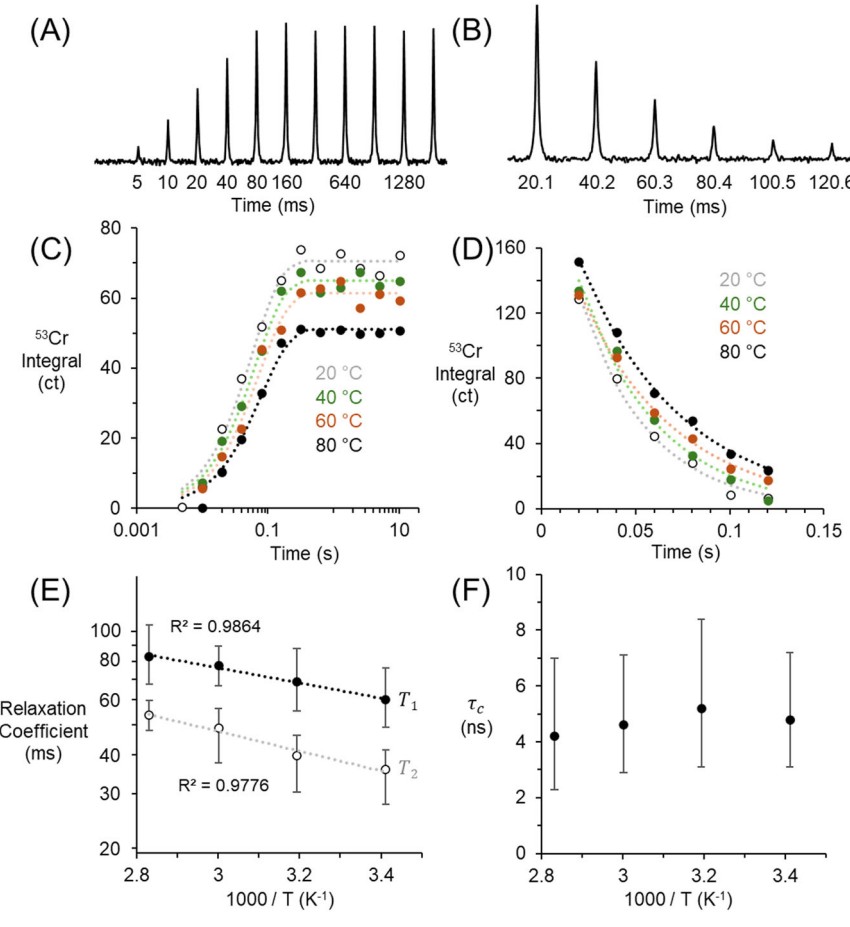

from that of the purely Gaussian distributions, thereby effecting the reported $D_T$. In contrast, the direct measurement of $D_T$ with PFGSTE NMR spectroscopy is well modeled by a symmetric Gaussian distribution. These skewed distributions were found for $T_1$, $T_2$, $\tau_c$ and the resulting $D_T$ across the temperature range between 20 and 80 °C, as shown in Figs. S2–S5. In general, the deviations from Gaussian uncertainty are more apparent for $T_1$ and $T_2$ than $\tau_c$ and $D_T$.

The consequences of these observations provided by Monte Carlo simulations of uncertainty are twofold. While the $T_1$ and $T_2$ method offers relatively precise diffusivity coefficients and efficient use of instrument time, its accuracy was validated using PFGSTE NMR, which provides direct measurements of translational diffusivity. Given that the PFG NMR experiments exhibited symmetric error, while the $T_1$ and $T_2$ measurements did not, the Monte Carlo approach provided an agnostic means to capture these asymmetries and better assess the uncertainty in the data. Capturing uncertainties associated with the $T_1$ and $T_2$ measurements evidently benefit from methodologies that can detect and quantify the asymmetry of the uncertainty and how these asymmetries propagate through BPP and subsequent equations, as most analytic approximations of uncertainty assume symmetric errors.

Secondly, further analysis of the sensitivity of fit parameter uncertainty to experimental methodology facilitates both a-priori determination of optimum parameters and the tradeoffs under practical conditions, with a speculated extension to autonomous, self-driven experiments. Such experiments could weigh the benefit of increasing the signal-to-noise ratio by increasing the number of transients versus increasing the number of increments (or sampling density) in gradient strength, potentially finding an optimal solution that balances the uncertainty associated with a measurement with the total instrument time.

The application of Monte Carlo simulations of uncertainty is distinct but synergistic to further advancement of analytical technologies

that offer exciting opportunities in PFGSTE NMR spectroscopy. For instance, stronger gradient probes as well as amplifiers can provide more powerful magnetic field gradients, improving the resolution and accuracy of diffusion measurements, especially for nuclei with low gyromagnetic ratios like $^{53}$Cr[52]. The use of cryoprobes can enhance the sensitivity of NMR measurements by significantly reducing thermal noise with some recent application to diffusion measurements via NMR spectroscopy[53]. Ultrafast Laplace NMR techniques that employ spatial encoding significantly reduce experiment times, enabling single-scan measurements and enhancing sensitivity through hyperpolarization techniques such as dissolution dynamic nuclear polarization[54]. These combined technologies could significantly advance the field by enabling efficient analysis of traditionally challenging NMR-active nuclei. Further improvements in sensitivity are especially true given that the maximum contaminant level for total chromium in groundwater as defined by the United States Environmental Protection Agency is 0.1 mg/L[55].

This study, in summary, demonstrates the feasibility of detecting low concentrations of $^{53}$Cr in multicomponent simulant nuclear waste. The high linearity with respect to CrO$_4^{2-}$ concentration despite changes to the composition of multicomponent concentrated electrolytes indicates that the integral is robust and independent of variations in the matrix composition, presenting an avenue for reliable quantification under diverse conditions. Additionally, this work indicates that technology is on the cusp of reliably obtaining diffusion data for $^{53}$Cr using PFGSTE NMR spectroscopy, despite inherent challenges related to the small gyromagnetic ratio and experimental constraints. These advancements, in tandem with further use of Monte Carlo simulations to optimize experimental protocols, could provide a detailed description of transport properties, facilitating the development of computational models and simulations that predict the behavior of untraditional NMR-active nuclei in complex environmental and industrial matrices.

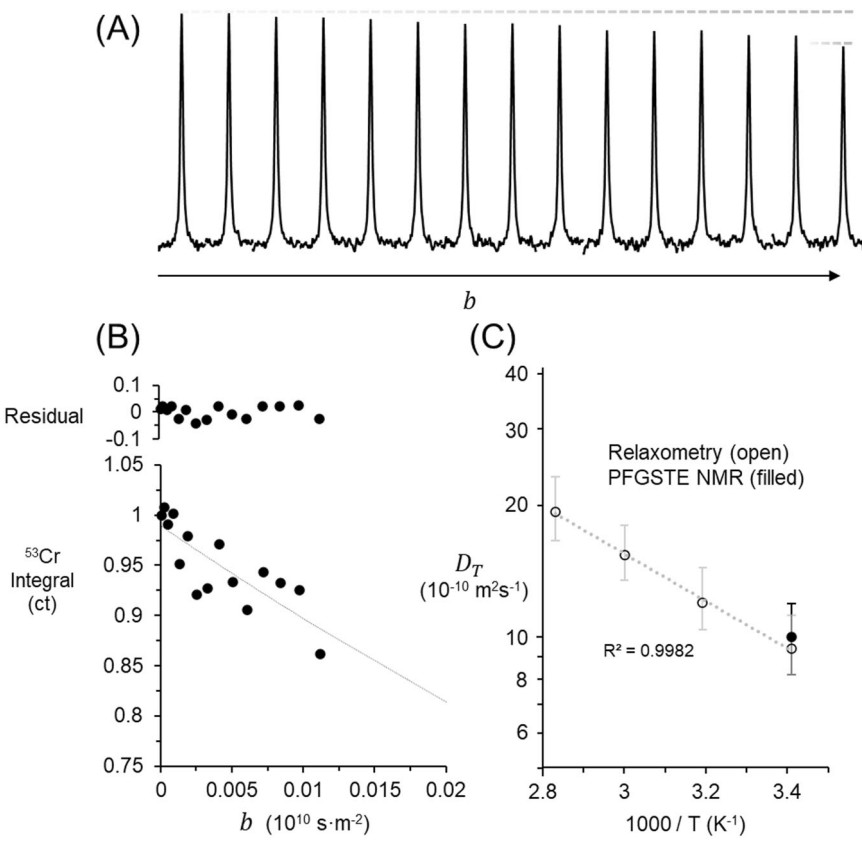

**Fig. 3 | Diffusion analysis of 1.47 m KOH and 1.75 m K$_2$CrO$_4$ solutions using $^{53}$Cr NMR. A** $^{53}$Cr PFGSTE NMR spectra of 1.47 m KOH and 1.75 m K$_2$CrO$_4$ at 20 °C. **B** Stejskal Tanner plot in which the logarithm of the normalized signal intensity is related to b, for instrument parameters contributing to beta, see the methodology section. **C** Temperature dependence of the $D_T$ estimated with BPP, Stokes-Einstein-Debye and Stokes-Einstein Equations with comparison the direct measurement of $D_T$ at 20 °C with PFGSTE NMR. A table of $D_T$ is provided in Table S7 of the *Supplementary Information*. The error bars (+/- σ) in this figure were determined via analysis of Monte Carlo simulations as later described. Note, the units of concentration are in molality.

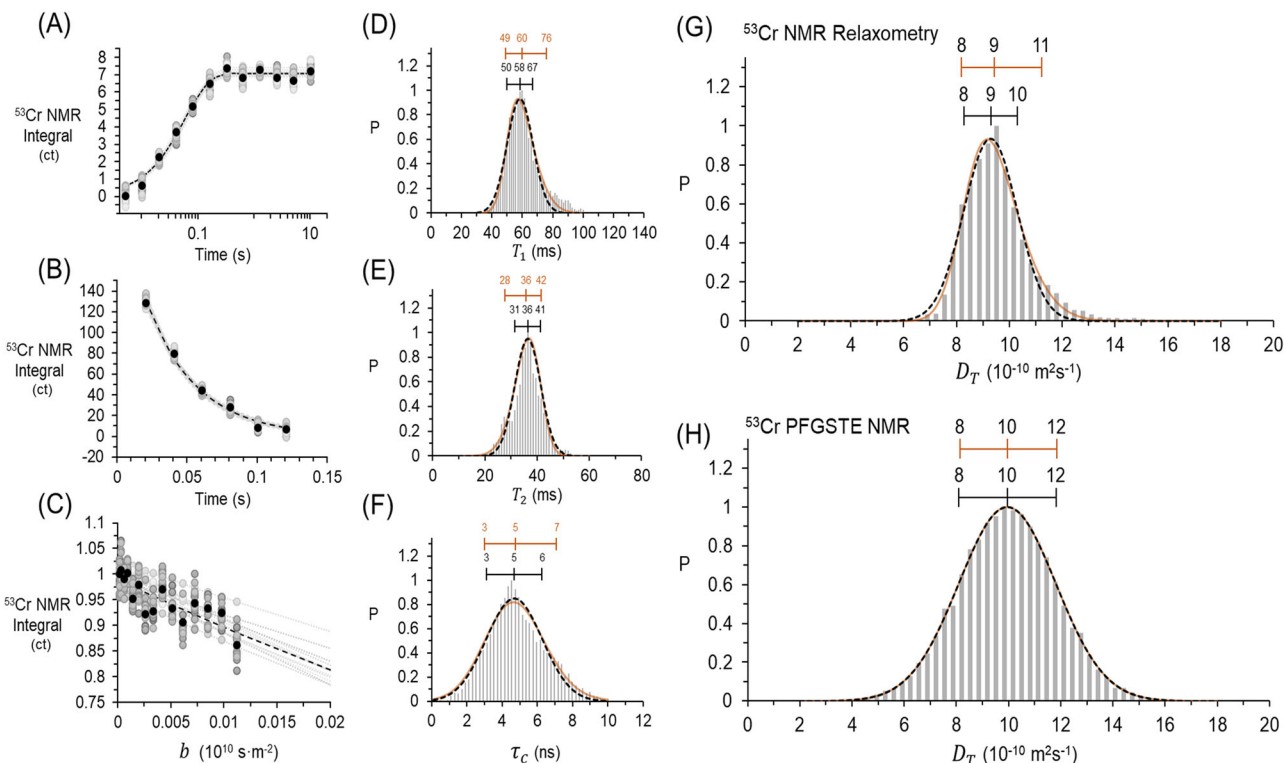

**Fig. 4 | Monte Carlo uncertainty analysis of NMR relaxometry and diffusion experiments.** Analysis (**A**) Inversion recovery experiments where 20 of 10,000 simulations are shown (**B**) CPMG experiments, where 20 of 10,000 simulations are shown (**C**) PFGSTE NMR experiments, where 20 of 14,000 simulations are shown. Resulting distributions of (**D**) $T_1$, (**E**) $T_2$, (**F**) $\tau_c$, (**G**) $D_T$ estimated from relaxometry (**H**) $D_t$ measured from $^{53}$Cr PFGSTE NMR, where P is probability. Skewed Gaussian fits are in orange and symmetric gaussian fits are marked with a black dashed line. Uncertainties (+/− σ) are likewise annotated.

## Methods

### Sample preparation

Multicomponent Hanford simulants were prepared as described in Schonewell et al.[40], and comprised chromate $[CrO_4^{2-}]$, aluminate $[Al(OH)_4^-]$, oxalate $[C_2O_4^{2-}]$, carbonate $[CO_3^{2-}]$, sulfate $[SO_4^{2-}]$, phosphate $[PO_4^{3-}]$, nitrite $[NO_2^-]$, nitrate $[NO_3^-]$, chloride $[Cl^-]$, fluoride $[F^-]$, hydroxide $[OH^-]$, calcium $[Ca^{2+}]$, cesium $[Cs^+]$, potassium $[K^+]$, sodium $[Na^+]$, strontium $[Sr^{2+}]$, acetate $[CH_3CO_2^-]$, and formate $[CHO_2^-]$. As detailed elsewhere[40], Inductively Coupled Plasma-Optical Emission Spectrometry (ICP-OES) was used to quantify simulant loadings of Al, Ca, Cr, Na, and K. Inductively coupled plasma mass spectrometry (ICP-MS) was then used to measure the Cs and Sr content of the simulant. Simulant anion loadings $(Cl^-, F^-, NO_3^-, NO_2^-, PO_4^{3-},$ and $SO_4^{2-})$ were next quantified using Ion Chromatography. A table of the concentrations is included in the *Supplementary Information*.

For the idealized solution of $CrO_4^{2-}$, potassium salts were chosen for their relatively low viscosity compared to equimolal solutions of sodium hydroxide[49,56]. Specifically, potassium dichromate ($K_2Cr_2O_7$, Sigma Aldrich, ACS Reagent Grade) was added to produce $CrO_4^{2-}$ in alkaline solutions. A concentrated potassium hydroxide (KOH, Sigma Aldrich, ACS Reagent Grade) solution was also prepared. To minimize oxygen contamination, deoxygenated water (18 MΩ·cm) was used, which was prepared by purging nitrogen ($N_2$) through boiling water within an $N_2$-filled glovebox overnight. The mixing procedure involved dissolving the potassium hydroxide salt in deoxygenated water, followed by addition of $K_2Cr_2O_7$. The potassium hydroxide chromate solution corresponded to 1.47 m KOH and 1.75 m $K_2CrO_4$, where m is in units of molality (moles/kg$_{water}$).

### NMR spectroscopy

Chromium-53 NMR spectroscopy was conducted using an 11.7534 T NMR spectrometer (Avance III, Bruker) equipped with a Bruker BBO SmartProbe and a GAB gradient amplifier capable of generating up to 50 G/cm. Temperature calibration was achieved using the $^1H$ chemical shifts of a flame-sealed ethylene glycol sample[57]. Single-pulse, direct excitation $^{53}Cr$ NMR spectra were collected with a time domain size of 590 points, a sweep width of 40.1935 ppm, and an acquisition time of 259.6 ms. A $\pi/2$ pulse width corresponding with 50 µs was applied, calibrated to 1.47 m KOH and 1.75 m $K_2CrO_4$, and the chemical shift (0 ppm) was also referenced to that concentrated solution. The relaxation delay was set to 0.0391 s, and 262144 scans performed, resulting in a total experiment time of 1 day. The spectra were analyzed in Mestrenova, zero-filled to 4096 complex points, and a 5 Hz exponential window function was applied.

To measure the $^{53}Cr$ $T_1$ relaxation times, a saturation recovery experiment (satrect1) was employed. The time domain size was 1136 points, with a sweep width of 40.19 ppm and an acquisition time of 499.8 ms. The recycle delay between scans was 0.5 s, and the saturation pulse train consisted of 16 pulses, each separated by 10 ms. The $\pi/2$ pulse width was calibrated at each temperature. Sixteen dummy scans were utilized, and 256 scans acquired for each of the 13 saturation delay steps (ranging from 0.005 s to 20.48 s). This setup resulted in a 2.5 h experiment.

The $T_2$ relaxation times for $^{53}Cr$ were determined using the Carr-Purcell-Meiboom-Gill (CPMG) sequence. The time domain size was 1136 points, with a sweep width of 40.19 ppm and an acquisition time of 499.8 ms. The recycle delay between scans was 0.5 s. The $\pi/2$ pulse width was calibrated at each temperature. Sixteen dummy scans were utilized, the pulse delay steps ranged from 20.12 ms to 181.04 ms, and the number of transients collected at each step was 768, resulting in a total experiment time of 1.5 h. The $T_1$ and $T_2$ measurements were processed in Mestrenova, where the spectra were zero-filled to 2048 complex points and then 5 Hz of exponential line broadening was applied.

For diffusion measurements, PFGSTE $^{53}Cr$ NMR spectra were recorded using a 2D sequence with convection compensation and LED bipolar gradients. The time domain size was 1470 points, with a sweep width of 9.9965 ppm and an acquisition time of 646.8 ms. The diffusion time was set to 70 ms, the spoiler gradient pulse was 600 µs, and the diffusion encoding gradient pulse ($\delta/2$) was 4450 µs. It is important to ensure gradient pulse durations and delays comply with safe operating conditions for probe and hardware, as specified in their manuals. Sixteen gradient steps (SMSQ10.100) were used, with a 0.92 s delay between scans. The recycle delay plus acquisition time totaled 1 s, significantly longer than $5 \cdot T_1$. This, combined with a 70 ms diffusion time, reduced the work duty cycle of the pulsed field gradients. Each gradient step involved 2176 scans at 20 °C, resulting in an experiment time of 16 hours. This experiment was then repeated a total of 20 times in succession on the same sample resulting in a total experiment time of 2 weeks. Arranging the measurement in this manner lowered the consecutive duration in which the highest gradient strengths were applied. The gradient powers are referenced to 1% $H_2O$ in $D_2O$ with 0.1% $CuSO_4$ at 25 °C (D = $19.1 \cdot 10^{-10}$ m$^2$s$^{-1}$).

### Monte Carlo simulations

The uncertainty associated with the data was explored via Monte Carlo simulations based on the analysis of the standard deviation of the residuals. Residuals were obtained by subtracting the observed data from a fitted baseline model, and their standard deviation on a Gaussian basis was calculated to estimate the variability in the data. Using this Gaussian standard deviation as a measure of uncertainty, a random number generator was then used to propagate the data and simulate new data points. Assumptions were made about the Gaussian noise model in our analysis, which assumed that the noise had a mean of 0 (no bias) and, as stated above, that the standard deviation of the noise was that of the residuals. Noise was then added directly in the frequency domain. This method thereby allowed for addition of Gaussian noise in the frequency domain with variance matching the residuals between the observed data and the fitted model. The simulated data (10,000 and 14,000 simulations for NMR relaxation measurements and PFGSTE NMR measurements, respectively) generated from the Monte Carlo simulation were analyzed using both Gaussian and skewed Gaussian fitting techniques to determine distributions of $T_1$, $T_2$, $\tau_c$, and subsequently $^{53}Cr$ self-diffusion coefficient ($D_t$) by both $^{53}Cr$ NMR relaxometry as well as $^{53}Cr$ PFGSTE NMR. Histograms of these parameters associated with each simulation were compiled and fit with a Gaussian distribution and a skewed Gaussian distribution.

## Data availability

Data are available upon reasonable request.

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

## Acknowledgements

This research was supported by IDREAM (Ion Dynamics in Radioactive Environments and Materials), an Energy Frontier Research Center funded by the U.S. Department of Energy (DOE), Office of Science, Basic Energy Science (BES) under FWP 68932. Pacific Northwest National Laboratory (PNNL) is a multiprogram national laboratory operated for DOE by Battelle Memorial Institute operating under Contract No. DE AC05-76RL0-1830. Graduate Fellows, A.R.K. and J.M., were supported by the Department of Energy Office of Environmental Management—Minority Serving Institutions Partnership Program (EM MSIPP). Carolyne Burns (PNNL) is thanked for generously providing the multicomponent Hanford waste simulants. Kee Sung Han (PNNL) and Micah Prange (PNNL) are thanked for helpful discussions.

## Author contributions

T.R.G. ideation, experimental methodology, data acquisition, data analysis, data interpretation, wrote initial draft, revisions. A.R.K. experimental methodology, J.M. experimental methodology, J.G.R. interpretation. C.I.P. supervision and funding acquisition. All authors have given approval to the final version of the manuscript.

## Competing interests

The authors declare no competing interests.
