## [Peer review file · Communications Chemistry]

Determining Hexavalent Chromium Transport Properties in Alkaline Nuclear Waste Using Nuclear Magnetic Resonance Spectroscopy

Corresponding Author: Dr Trent Graham

Version 0:

Reviewer comments:

Reviewer #1

(Remarks to the Author)

In this work, the authors study the feasibility of correlating the ^{53}Cr NMR signal with Cr(VI) concentration in multicomponent Hanford simulants and evaluating the Cr diffusion coefficient using relaxation times and the PFGSTE sequence. Monte Carlo simulations are performed to assess the uncertainties in the extracted quantities. Their results show a strong correlation between the ^{53}Cr NMR signal and Cr(VI) concentration, consistent diffusion coefficients obtained from the two methods, and provide interesting information on the uncertainties using Monte Carlo simulations.

The results are interesting and pave the way for a new, reliable method to evaluate Cr(VI) concentration and its diffusion properties. They appear sufficiently novel to warrant publication. The text is clear and provides enough information to reproduce the authors' results.

Thus, I recommend publication, provided that the following minor comments are addressed and the authors respond to the following questions:

Questions

- * In the Introduction, the authors state that Cr(VI) is a source of pollution. What is the tolerable Cr(VI) concentration in wastewater? Is it close to the detection limit of their proposed method?
- * It is not entirely clear how the translational diffusion coefficient is obtained from T1-T2. Did the authors follow this procedure: first determining τ_c from T1-T2, then calculating Rh using Equations (2)-(3) with the experimental values of η and T, and finally computing Dt using Equation (4)?
- * In Figure 2c, the NMR curves appear noisier at lower temperatures. Is there an explanation for this?
- * Figure 3b: The NMR integral is expected to follow an exponential decay (see Eq. (5)), yet the graph appears linear. Were the data points fitted using a linear or an exponential model?
- * Unless I missed it, Figures S2 to S5 are not referred in the main text. Could the authors comment on them and possibly provide a brief analysis of these results in the main text?

Minor Typos

- * Figure 2e: Does "relaxation coefficient" refer to relaxation times?
- * Page 3, second column, paragraph "In Equations 2-4...": It may not be necessary to explicitly define π .
- * Figure 3A: The x-axis lacks values and units.
- * Page 4, second column, paragraph "Figure 4A-B show...": "the resulting T₁ and T₂ were analyzed via BPP to determine the reorientail coefficient," the "reorientail" coefficient is not defined. I assume "reorientail" refers to the correlation time τ_c ?
- * Page 6, last paragraph: The sentence beginning with "CAUTION" does not seem to fit naturally into the text.

Reviewer #2

(Remarks to the Author)

This paper demonstrates that ^{53}Cr NMR can be used to quantify Cr(VI) to concentrations down to 3 mM, and that diffusion coefficients of these species can be measured by PFGSTE, in spite of low sensitivity and fast relaxation, thus encouraging exploration of other nuclei with similar unfavorable properties.

Question 1: What is the reason that saturation recovery and not inversion recovery was used for T1 measurement? In principle, the latter will have twice the precision of the first.

Question 2: In the PFGSTE experiments why was Delta not increased to produce a signal reduction of 80% instead of 20%? I expect that a larger signal decay would provide better precision for the diffusion coefficient.

Question 3: A thorough Monte-Carlo simulation is given for the analysis of the uncertainty in the diffusion coefficient predicted from T1 and T2. Do results in figure 4 G and H justify the conclusion that measurements of T1 and T2 of a total of 4 hrs provide a better result for the diffusion coefficient than a PFGSTE experiment of 16 hrs? If so, it should be mentioned in the text.

Except for figure 1c, the concentration is expressed in m (meters) instead of M (mole/liter).

Version 1:

Reviewer comments:

Reviewer #1

(Remarks to the Author)

The authors have carefully answered all my questions and have improved their manuscript. I therefore recommend publication.

Reviewer #2

(Remarks to the Author)

The reviewers' comments have been addressed thoroughly, therefore I recommend the manuscript for publication in the current form.

Reviewers' comments:

Reviewer #1 (Remarks to the Author):

Authors' responses (blue)

In this work, the authors study the feasibility of correlating the ^{53}Cr NMR signal with Cr(VI) concentration in multicomponent Hanford simulants and evaluating the Cr diffusion coefficient using relaxation times and the PFGSTE sequence. Monte Carlo simulations are performed to assess the uncertainties in the extracted quantities. Their results show a strong correlation between the ^{53}Cr NMR signal and Cr(VI) concentration, consistent diffusion coefficients obtained from the two methods, and provide interesting information on the uncertainties using Monte Carlo simulations.

The results are interesting and pave the way for a new, reliable method to evaluate Cr(VI) concentration and its diffusion properties. They appear sufficiently novel to warrant publication. The text is clear and provides enough information to reproduce the authors' results.

Thus, I recommend publication, provided that the following minor comments are addressed and the authors respond to the following questions:

Questions

1. In the Introduction, the authors state that Cr(VI) is a source of pollution. What is the tolerable Cr(VI) concentration in wastewater? Is it close to the detection limit of their proposed method?

We appreciate the reviewer's request for clarification on the tolerable concentration of Cr(VI) in wastewater and its relation to the detection limit of our proposed method. We have added discussion of this into the manuscript.

The EPA's maximum contaminant level for Cr(VI) in groundwater is 0.1 mg/L. To detect Cr(VI) at such low concentrations, significant advancements in our method's sensitivity or much longer total acquisition times would be needed. In contrast, the Cr(VI) concentration in Hanford tank waste is approximately 5 mM, which is considerably higher. Thus, our ^{53}Cr NMR methodology is currently more applicable for process monitoring in situations with higher concentrations of Cr(VI), such as industrial or waste processing environments, rather than for environmental monitoring purposes where lower detection limits are required.

Added text: Further improvements in sensitivity are especially true given that the maximum contaminant level for total chromium in ground water as defined by the United States, Environmental Protection agency is 0.1 mg/L.⁵⁴

54. Putra, N. R., Zaini, M. A. A., Kusuma, H. S., Darmokoesoemo, H. & Faizal, A. N. M. Advances in chromium removal using biomass-derived activated carbon: A comprehensive review and bibliometric analysis. *Environmental Progress & Sustainable Energy* n/a, e14598
[https://doi.org:https://doi.org/10.1002/ep.14598](https://doi.org/https://doi.org/10.1002/ep.14598)

2. It is not entirely clear how the translational diffusion coefficient is obtained from T1-T2. Did the authors follow this procedure: first determining τ_c from T1-T2, then calculating Rh using Equations (2)-(3) with the experimental values of η and T, and finally computing Dt using Equation (4)?

We thank the reviewer for highlighting the need for a clearer explanation of how the translational diffusion coefficient is derived from T1-T2 measurements. We have added text to clarify this procedure in the revised manuscript.

To determine the translational diffusion coefficient, we first measured the spin-lattice relaxation time (T1) and the spin-spin relaxation time (T2). Using these values, we calculated the molecular correlation time (τ_c). Subsequently, we used Equations (2) and (3), along with the experimental values for viscosity (η) and temperature (T), to compute the hydrodynamic radius (Rh). Finally, Dt was calculated using Equation (4).

These steps are now more clearly articulated in the manuscript to ensure that the methodology is transparent and easy to follow.

Added text: The relationship between T_1 and T_2 can then be used to estimate the rotational correlation time, (τ_c), for spin-3/2 nuclei (**Equation 1**),^{44 47,48} which describes the characteristic time scale of molecular motion. As shown in **Figure 2F**, τ_c was approximately constant across the temperature range, and τ_c was then used to calculate the self-diffusion coefficient (D_t) of ⁵³Cr using the Stokes-Einstein-Debye and Stokes-Einstein Equations (**Equation 2-4**).⁴³

3. In Figure 2c, the NMR curves appear noisier at lower temperatures. Is there an explanation for this?

We thank the reviewer for observing that the noise of the integrated signal in the saturation recovery T1 measurement varies as a function of temperature. The data were all collected with the same number of transients, and the absolute integral is shown. The integrated signal at low temperature is observed to increase, in agreement with expected trends from contributions of the quality factor (Q), Boltzmann factor, density, and other factors.

Regarding the noise, in Figure 2c, the data at 60°C have the highest normalized uncertainty of about 5%, followed by lower uncertainties at other temperatures. While we cannot completely isolate the contributions to noise, the magnitude of these uncertainties does not vary systematically with temperature or with d20.

We have added text to the manuscript to include this observation.

Added text: Regarding the noise, in Figure 2C, the data at 60°C have the highest normalized uncertainty of about 5%, followed by lower uncertainties at other temperatures.

4. Figure 3b: The NMR integral is expected to follow an exponential decay (see Eq. (5)), yet the graph appears linear. Were the data points fitted using a linear or an exponential model?

We thank the reviewer for their observation and have added text addressing this point in the manuscript. The fit function used for the data is exponential. However, the exponential fit appears quasi-linear across

the measurable range of b due to the limited dynamic range of signal attenuation, which is an intrinsic property of ^{53}Cr , specifically its low gyromagnetic ratio.

Stronger capabilities in probes and gradient amplifiers would enhance signal attenuation, making the exponential function more discernible and the measured diffusion coefficient more precise. To address the uncertainties that arise from this limited signal attenuation, we performed Monte Carlo simulations to rigorously estimate the uncertainty of the diffusion coefficient.

We have also added text to the manuscript to convey that quasi-linearity is observed, that this is attributed to the dynamic range of the attenuation arising from the intrinsic properties of ^{53}Cr , and that the limited dynamic range of the attenuation used to measure the diffusion coefficient was the rationale for the Monte Carlo simulations.

Added text: The exponential fit appears quasi-linear across the measurable range of b due to the limited dynamic range of signal attenuation.

5. Unless I missed it, Figures S2 to S5 are not referred in the main text. Could the authors comment on them and possibly provide a brief analysis of these results in the main text?

We thank the reviewer for encouraging a greater discussion of the supplementary figures (Figures S2 to S5) in the main text. These figures provide Monte Carlo results at different temperatures, estimating the error of T_1 , T_2 , τ_c , and D .

To address the reviewer's suggestion, we have referred to Figures S2 to S5 and provided a brief analysis of these results in the main text. This discussion includes how the Monte Carlo simulations were utilized to estimate uncertainties at various temperatures and the impact of these errors on our measurements of T_1 , T_2 , τ_c , and D . Additionally, we note that the uncertainties exhibit non-Gaussian characteristics, and these deviations are considered in our analysis.

Added text: These skewed distributions were found for T_1 , T_2 , τ_c and the resulting D_t across the temperature range between 20 and 80 °C, as shown in Figure S2-S5. In general, the deviations from Gaussian uncertainty are more apparent for T_1 and τ_c than T_2 and D_t .

6. Minor Typos

- Figure 2e: Does "relaxation coefficient" refer to relaxation times?
- Page 3, second column, paragraph "In Equations 2-4...": It may not be necessary to explicitly define π .
- Figure 3A: The x-axis lacks values and units.
- Page 4, second column, paragraph "Figure 4A-B show...": "the resulting T_1 and T_2 were analyzed via BPP to determine the reorientail coefficient," the "reorientail" coefficient is not defined. I assume "reorientail" refers to the correlation time τ_c ?
- Page 6, last paragraph: The sentence beginning with "CAUTION" does not seem to fit naturally into the text.

We thank the reviewer for pointing out these minor typos and suggestions. We have added minor edits to the text addressing their listed items.

Reviewer #2 (Remarks to the Author):

This paper demonstrates that ^{53}Cr NMR can be used to quantify Cr(VI) to concentrations down to 3 mM, and that diffusion coefficients of these species can be measured by PFGSTE, in spite of low sensitivity and fast relaxation, thus encouraging exploration of other nuclei with similar unfavorable properties.

11. What is the reason that saturation recovery and not inversion recovery was used for T1 measurement? In principle, the latter will have twice the precision of the first.

We appreciate the reviewer's question and have included a discussion of this in the manuscript.

Given the need to balance multiple types of NMR measurements across a range of temperatures, we opted for the saturation recovery experiments. This decision was made to optimize our overall experimental time and allow for a more extensive set of PFGSTE NMR experiments, as saturation recovery does not require the T1 value to be known a priori. This flexibility enabled more efficient data collection and allowed us to check for longer-than-expected T1 contributions without the need for preliminary measurements.

We have added the point raised by the reviewer and our rationale regarding saturation recovery and inversion recovery experiments to the manuscript to clarify our reasoning.

Added text: Note that in lieu of inversion recovery, saturation recovery experiments were chosen to reduce overall experimental time and allow for a more extensive set of PFGSTE NMR experiments, as saturation recovery does not require the T1 value to be known a priori.

12. In the PFGSTE experiments why was Delta not increased to produce a signal reduction of 80% instead of 20%? I expect that a larger signal decay would provide better precision for the diffusion coefficient.

We thank the reviewer for this question and have added text discussing this in the manuscript.

While increasing Delta in PFGSTE experiments to produce a larger signal reduction could potentially provide better precision for the diffusion coefficient, it would also lead to significant signal attenuation from T1 and T2 relaxation processes, dramatically decreasing signal intensity.

The limited dynamic range of signal attenuation is an intrinsic property observed in our study, attributed to the specific characteristics of the nuclei we are studying. Stronger capabilities in probes and gradient amplifiers would enhance signal attenuation, making the exponential function more discernible and the measured diffusion coefficient more precise.

We chose a Delta that offers a compromise between sufficient signal reduction for accurate diffusion coefficient estimation and maintaining signal intensity for reliable measurements. To further address any uncertainties arising from the limited signal attenuation, we performed Monte Carlo simulations to rigorously estimate the uncertainty of the diffusion coefficient.

We have added this rationale to the manuscript to clarify our reasoning.

Added text: Due to the small gyromagnetic ratio of ^{53}Cr , stronger gradients are needed to spatially resolve the spins. Whereas Δ can also be typically increased to improve the degree of signal attenuation, this is constrained by signal loss via T_1 and T_2 processes.

13. A thorough Monte-Carlo simulation is given for the analysis of the uncertainty in the diffusion coefficient predicted from T1 and T2. Do results in figure 4 G and H justify the conclusion that measurements of T1 and T2 of a total of 4 hrs provide a better result for the diffusion coefficient than a PFGSTE experiment of 16 hrs? If so, it should be mentioned in the text.

We thank the reviewer for this insightful question and have included a discussion of this point in the manuscript.

While the T1/T2-derived method offers higher precision of translational diffusivity coefficients, validating these results with PFGSTE NMR ascertained their accuracy. In addition, PFGSTE NMR also provides a direct measurement of translational diffusivity, unlike T1/T2-based calculations which rely on Stokes-Einstein assumptions. With the current instrumental setup, T1 and T2 measurements are a more efficient use of instrument time to estimate diffusivity coefficients. The acquisition of a probe and gradient amplifier capable of stronger pulsed field gradients would facilitate a reduction in the diffusion delay, improving the sensitivity of the measurement and reducing acquisition time, in addition to increasing the extent of signal reduction also improving the accuracy of the diffusion coefficient measurement.

We have added this rationale and comparison to the manuscript for clarity.

Added text: While the T1/T2 method offers relatively precise diffusivity coefficients and efficient use of instrument time, its accuracy was validated using PFGSTE NMR, which provides direct measurements of translational diffusivity.

14. Except for figure 1c, the concentration is expressed in m (meters) instead of M (mole/liter).

We thank the reviewer for pointing out the need to clarify the units of concentration. We have revised the manuscript to highlight the units of concentration more clearly.

Figure 1c uses molarity (M, mole/L) for concentration, while the rest of the manuscript uses molality (m, moles/kg water) because molality is temperature invariant, whereas molarity varies with temperature due to changes in the volume and density of the solution.

Added text: Note, the units of concentration are in molality.

REVIEWERS' COMMENTS:

Reviewer #1 (Remarks to the Author):

The authors have carefully answered all my questions and have improved their manuscript. I therefore recommend publication.

Reviewer #2 (Remarks to the Author):

The reviewers' comments have been addressed thoroughly, therefore I recommend the manuscript for publication in the current form.

Authors' response: The authors thank the two reviewers for their constructive feedback throughout the manuscript process.